# Gentle Touch Therapy, Pain Relief and Neuroplasticity at Baseline in Fibromyalgia Syndrome: A Randomized, Multicenter Trial with Six-Month Follow-Up

**DOI:** 10.3390/jcm11164898

**Published:** 2022-08-20

**Authors:** Afonso Shiguemi Inoue Salgado, Miriam Hatsue Takemoto, Carla Fernanda Tallarico Carvalho de Souza, Daiana Cristina Salm, Danielli da Rosa, Gabriela Correa Cardoso, Daniela Dero Ludtke, Silvia Fiorillo Cabrera Soares, Júlia Koerich Ferreira, Aline Raulino Dutra, Yuri Cordeiro Szeremeta, Gustavo Mazzardo, Joice Mayra, Débora da Luz Sheffer, Wolnei Caumo, Edsel B. Bittencourt, Robert Schleip, Alexandra Latini, Franciane Bobinski, Daniel Fernandes Martins

**Affiliations:** 1Natural Quanta Wellness Center, Windermere, FL 32835, USA; 2Experimental Neuroscience Laboratory (LaNEx), University of Southern Santa Catarina, Palhoça 88137-272, Brazil; 3Postgraduate Program in Health Sciences, University of Southern Santa Catarina, Palhoça 88137-272, Brazil; 4Laboratório de Bioenergética e Estresse Oxidativo, Departamento de Bioquímica, Centro de Ciências Biológicas, Universidade Federal de Santa Catarina, Florianópolis 88040-900, Brazil; 5Department of Anatomy, Institute of Biomedical Sciences, University of São Paulo, São Paulo 05508-900, Brazil; 6Post-Graduate Program in Medical Sciences, School of Medicine, Universidade Federal do Rio Grande do Sul (UFRGS), Porto Alegre 90010-150, Brazil; 7Laboratory of Pain and Neuromodulation, Hospital de Clínicas de Porto Alegre (HCPA), Porto Alegre 90035-903, Brazil; 8Laboratory of Neuromodulation and Center for Clinical Research Learning, Physics and Rehabilitation Department, Spaulding Rehabilitation Hospital, Boston, MA 02129, USA; 9Coastal Health Institute, Jacksonville, FL 32224, USA; 10Department of Sport and Health Sciences, Technical University of Munich, 80799 Munich, Germany; 11Department for Medical Professions, DIPLOMA University of Applied Sciences, 37242 Bad Sooden-Allendorf, Germany

**Keywords:** chronic pain, inflammatory biomarkers, manual therapy, neurotransmitter systems, osteopathic treatment

## Abstract

Background: Fibromyalgia (FM) is considered a stress-related disorder characterized mainly by chronic widespread pain. Its pathogenesis is unknown, but cumulative evidence points at dysfunctional transmitter systems and inflammatory biomarkers that may underlie the major symptoms of the condition. This study aimed to evaluate pain scores (primary outcome), quality of life, inflammatory biomarkers and neurotransmitter systems in women with FM (secondary outcomes) subjected to gentle touch therapy (GTT) or placebo. Methods: A total of 64 female patients with FM were randomly assigned to two groups, namely GTT (n = 32) or Placebo (n = 32). Clinical assessments were conducted at baseline and post-intervention with six-month follow-up. We measured serum catecholamines (dopamine), indolamines and intermediary metabolites (serotonin or 5-hydroxyindolacetic acid (5-HIAA)), as well as tetrahydrobiopterin (BH4), which is a cofactor for the synthesis of neurotransmitters and inflammatory biomarkers in women with FM. A group of healthy individuals with no intervention (control group) was used to compare biochemical measurements. Intervention effects were analyzed using repeated measures (RM) two-way ANOVA followed by Bonferroni post hoc test and mixed ANCOVA model with intention to treat. Results: Compared to placebo, the GTT group presented lower pain scores and brain-derived neurotrophic factor (BDNF) levels without altering the quality of life of women with FM. Changes in BDNF had a mediating role in pain. Higher baseline serum BDNF and 5-HIAA or those with a history of anxiety disorder showed a higher reduction in pain scores across time. However, women with higher serum dopamine levels at baseline showed a lower effect of the intervention across the observation period revealed by an ANCOVA mixed model. Conclusions: In conclusion, lower pain scores were observed in the GTT group compared to the placebo group without altering the quality of life in women with FM. Reductions in BDNF levels could be a mechanism of FM pain status improvement. In this sense, the present study encourages the use of these GTT techniques as an integrative and complementary treatment of FM.

## 1. Introduction

Fibromyalgia (FM) is a highly prevalent, disabling, multifactorial syndrome in which individuals experience pain, increased emotional stress, sleep disturbances, depressed mood, fatigue, catastrophic thinking and memory impairment [1,2]. Although some etiological factors have been proposed, such as genetics, physical or emotional trauma, it is still considered a syndrome of undefined cause [3,4]. Because a pattern of pathological, imaging or biochemical features has not yet been characterized, the diagnosis of FM is still uncertain. However, the study of different biomarkers in this syndrome has been justified by the involvement of the neuro-immune axis [5]. It has been demonstrated that patients with FM have alterations in the neurotransmitter systems, which may be a neurobiological explanation of the main symptoms of this disease [6,7]. Pain has been described as the most frequent and disabling symptom in individuals with FM [8], with a significant impact on functional capacity and quality of life [9]. It is classified as nociplastic pain, that is, when there is an abnormal central processing of pain [10].

Derangements in neurotransmitters and neuromodulators such as serotonin, norepinephrine and dopamine are important in endogenous pain inhibition pathways and are altered in FM in a way that could account for the increased pain experienced by the individual [6,7]. Concentrations of serotonin and noradrenaline appear to be lowered in the central nervous system [11,12], possibly contributing to dysfunctional descending pathways [13]. Likewise, dopamine activity has been shown to be attenuated in FM [14,15]. Besides the pain sensitivity, alterations in noradrenaline and serotonin may also contribute to disorder in mood with a deficient serotonin system strongly associated with major depression in FM [16].

Central nervous system (CNS) mechanisms such as central sensitization (CS), neural facilitation and disinhibition are involved in dysfunctional pain (widespread allodynia and hyperalgesia) and are found in individuals with FM [17]. The elevated serum levels of brain-derived neurotrophic factor (BDNF) observed in FM individuals correlate with lower pain thresholds [18] and are commonly identified in the CS process [19]. BDNF also plays a crucial role in neuroplasticity changes and therefore in the onset and evolution of pain at the CS level. In the pain pathway, BDNF weakens GABAergic synapses while strengthening glutamatergic synapses [20,21,22], acting as a regulator of neuronal activity and neuronal plasticity dependent on the N-methyl-D-aspartate (NMDA) receptor [20]. The main effect of this neurotrophic factor is mediated by the polarity shift of GABA concentrations in dorsal horn neurons [23]; thus, the GABAergic system loses the ability to downregulate the expression of the K^+^-Cl-exporting Cl-cotransporter (KCC2) in the dorsal horn [21,22]. Subsequently, the intracellular increase of Cl^−^ induces the loss of the GABAergic inhibitory effect on these nociceptors, which leads to disinhibition [21].

It is a well-described phenomenon that the activation of pattern recognition receptors present on peripheral immune cells during the inflammatory process contributes to pain [24,25,26]. This provides evidence for the role of immune cells in peripheral sensitization [22]. Furthermore, receptors for many inflammatory cytokines are expressed on dorsal horn neurons [24], which, when activated, can increase excitation and suppress synaptic inhibition in second-order neurons in the spinal cord [20]. On the other hand, central nervous system activity appears to modulate immune activity, but how cytokines induce central sensitization is still unclear [27,28]. Increased concentrations of interleukin 8 (IL-8) play a role in pain induction and maintenance [29] in FM [30], which can lead to poor quality of life [31].

Mendieta et al. (2016) found higher concentrations of interleukin 8 (IL-8) in FM patients vs. healthy volunteers and this was correlated with clinical pain scores, suggesting that IL-8 has chronic pain maintenance effects in individuals with FM [32]. IL-8 promotes sympathetic pain by stimulating production and altering the hypothalamic–pituitary (HPA) axis, supporting its potential importance in FM syndrome [33,34]. In addition, Uçeyler et al. (2011) have shown lowered IL-10 relative gene expression and protein levels in 26 individuals with FM as defined by the criteria of the American College of Rheumatology (ACR) [35].

Pro-inflammatory and pro-oxidant mediators are known to rapidly activate tetrahydrobiopterin (BH4) synthesis by upregulating *GCH1*, which encodes for the rate-limiting enzyme of the de novo BH4 pathway, GTP cyclohydrolase (GTPCH) [36,37]. BH4 is traditionally defined as an obligate cofactor for the synthesis of the neurotransmitters serotonin and dopamine, nitric oxide, as well as for the metabolism of phenylalanine and lipid ethers [36,37]. However, it has recently been shown that increased levels of BH4 are pro-nociceptive and the use of specific inhibitors for the synthesis of this pterin induces analgesia [38]. Correlated with low neuropathic pain scores, the discovery of single-nucleotide polymorphisms at the GCH1 loci was the first human validation linking BH4 to chronic pain [39]. Human homozygous haplotypes were associated with decreased upregulation of *GCH1*, protein and final metabolites upon stimulation in human white blood cells [39]. It was later demonstrated that homozygous carriers of this pain-protective haplotype [39] are also less sensitive to persistent pain in FM [40]. Additionally, a positive correlation of increased BH4 metabolism with pain scores has been reported in patients affected by complex regional pain and by diabetic neuropathic pain [41,42]. However, to the best of our knowledge, no previous study has investigated the levels of BH4 in biological fluids in FM-affected individuals. Considering that the metabolism of BH4 can be assessed in the blood and in the urine, the identification of these potential biomarkers might offer relevant information about the FM progression and pathogenesis.

Manual therapy (MT) is a non-pharmacological treatment provided by chiropractors, physical therapists and osteopaths, among other healthcare professionals, conceptualized as the treatment of dysfunctions in muscles, tendons, ligaments, joints, nerves, skin and organs performed by the hands of a therapist, covering a variety of refined techniques that aim to mobilize or manipulate the soft tissues of these structures [43,44]. The primary objective of MT is to reduce pain and increase range of motion and function [43,44]. In animal models, there is scientific support regarding the effectiveness of manual therapy for regulating nerve injury pain [45], post-operative pain [46,47,48], and Complex Regional Pain Syndrome Type I [49] with joint mobilization. In humans, osteopathic manipulative treatment (OMT) on cardiac autonomic modulation in healthy individuals causes a significant increase in parasympathetic activity and a decrease in sympathetic activity [50]. There are several MT techniques, including the GTT technique. During treatment, touch can be performed on different layers of the human body, such as fascia, muscles, tendons, ligaments, bones, viscera and nerves. However, manual treatment is mostly performed on the skin (epidermis) [51,52,53]. Mobility tests of these structures are then carried out using both hands of the therapist, who selectively checks if there is a decrease in the physiological movements of each structure tested. The advantages of gentle manual therapy in relation to traditional MT are as follows: it does not present contraindications due to the softness of its therapeutic gestures on the body tissues, it demands less (physical) energy expenditure for the therapist and for the patient during the session of treatment, among others [51,52,53]. There is promising evidence of effectiveness of GTT in diverse chronic pain conditions including FM [51], inflammatory-related chronic conditions such as irritable bowel syndrome [52] and nociceptive pain such as post-traumatic acute neck pain [53]. Interestingly, alterations in levels of pro- and anti-inflammatory cytokines have been reported after a GTT intervention in a non-clinical study on acute stress induced by the sleep deprivation model [54]. However, no study has evaluated the effects of GTT intervention on neuro-inflammatory biomarkers in patients with FM.

Considering all the theoretical framework mentioned, the clinical and biochemical characteristics of FM and in addition all the known benefits of GTT and MT techniques [45,46,47,48,49,50,51,52,53,54], in this study characterized as a randomized controlled clinical trial, we compared two sessions of GTT 45 days apart with placebo intervention on pain scores (primary outcome). In addition, we examined the impact of the cumulative GTT on pain at the 3- and 6-month follow-up. Secondary outcomes were quantification of quality of life, inflammatory biomarkers (IL-8 and IL-10) and neurotransmitters (Serotonin, 5-hydroxyindolacetic acid (5-HIAA), tetrahydrobiopterin (BH4) and dopamine). Additionally, we investigated whether neuroplasticity status, assessed by serum BDNF and other neurotransmitters at baseline, could predict the lowest pain scores.

## 2. Materials and Methods

### 2.1. Study Design and Eligibility

This randomized, double-blinded, multicenter, parallel-group trial compared 1 intervention protocol of MT for the management of FM: GTT versus placebo. The impact of fibromyalgia on participants was evaluated at baseline and 15, 45, 60, 90, 120 and 180 days after interventions (Figure 1). Both patients and investigators were blinded to group assignments. In addition, a group of healthy individuals with no intervention (control group) was used to compare biochemical measurements between FM and healthy individuals.

The primary outcome was rated pain on the Visual Analogue Scale (VAS) for Pain. Secondary outcomes included the short form of the McGill Pain Questionnaire (MPQ-SF) for Pain, health status as measured by the Fibromyalgia Impact Questionnaire (FIQ) and Short-Form Health Survey (SF-36). Neurotransmitter systems and inflammation were measured by the serum BDNF, IL-10, IL-8, dopamine, serotonin, 5-hydroxyindolacetic acid (5-HIAA) and BH4 concentration analysis.

The clinical trial was conducted following the Consolidated Standards of Reporting Trials (CONSORT) extension for nonpharmacologic treatments (NPTs) [55]. The study was approved by the Research Ethics Committee at UNISUL, Santa Catarina, Brazil (CAAE: 03393318.0.0000.5369) and the trial was registered in the Brazilian Registry of Clinical Trials (ReBEC) (ensaiosclinicos.gov.br: RBR-2p8rrh) with Universal Trial Number (UNT: U1111-1225-8036) and was carried out in accordance with the Declaration of Helsinki [56]. We obtained oral and written informed consent from all patients before participating in this study.

### 2.2. Inclusion and Exclusion Criteria

Patient selection occurred at the Physical Therapy Clinic of the University of Southern Santa Catarina (UniSul), which is a private teaching Medical School located in the South of Brazil and private clinics in the cities of Pinhão, Paraná-PR and Guarapuava, PR. All evaluations and treatments were performed at the Physical Therapy Clinic of the UniSul and private clinics in the PR. Initially, patients were invited to answer a questionnaire to assess if they met inclusion criteria. Sixty-four women diagnosed with FM were recruited from February 2019 to March 2020 to participate in this study. Eighteen healthy subjects (control group), mean age 54.3 ± 9.4 years and body mass index (BMI) 27.9 ± 6.4 kg/m^2^) were selected from the community to compare biochemical measurements between FM and healthy individuals. However, they did not receive any intervention. A structured questionnaire screened them to ensure that they had no significant health-related issues and had the absence of any acute or chronic disease or medication use. Inclusion criteria for the participants were: (i) women between 30–75 years old, an age range in which FM becomes more prevalent, (ii) diagnosed according to the 2016 American College of Rheumatology criteria for FM [57] and (iii) having received pharmacological treatment for more than three months with no clinical improvement. Exclusion criteria were: (i) intake of anti-inflammatory drugs or drugs that affect the inflammatory process in the last month, (ii) known neurological disorder, (iii) peripheral neuropathy, (iv) pregnancy or breast-feeding, (v) known serious cardiovascular disease (i.e., uncontrolled arterial hypertension, cardiac pacemaker), (vi) neoplasia, (vii) surgery in the last four months, (viii) use of psychoactive drugs or narcotics. Current levels of activity were advised (physiotherapy, water exercises, acupuncture, etc.), with no new regular activity permitted.

### 2.3. Sample Size Justification

The sample size estimation was established using a 2-tailed hypothesis to detect 1.2 cm test and a standard deviation of 2.5 for cumulative scores of the pain scores on the VAS at six follow-up assessments for a randomized ratio of 1:1 to receive GTT vs. placebo, simulated for type I error of 5% and type II error of 20% (assuming a power of 0.80, the two-sided, significance level of 0.05, 0.5 correlation among repeated measures). The analysis indicated a sample size of 50 subjects, divided into two balanced groups (n = 50). Considering a dropout of 20% across the study’s time, the final number of patients was 64 patients (32 per group). Considering that the dropout was 26.5%, a post-hoc sample size calculation revealed that a sample of 37 patients had a cumulative difference in the absolute change in pain scores equal to 16.60% (standard deviation equal to 26.4). For the effect size (f2) equal to 0.62, the power of analysis was 0.94, and type I and II errors were 0.05 and 0.20, respectively.

### 2.4. Randomization

The participants were randomized by the Research Randomizer version 4.0, available at http://www.randomizer.org/ (accessed on 10 January 2019), by an external assistant who was blinded to the study objectives to assign 64 patients to an allocation of 1:1 for GTT or placebo groups. To avoid predicting the next patient, we used randomization in six blocks of eight. Before the recruitment phase, two investigators not involved in the patient’s assessments made the randomization. They prepared the envelopes that contained the randomization number. These envelopes were sealed, numbered sequentially, and they were opened after the participant consented to participate in the trial according to the numerical order registered outside.

### 2.5. Blinding

Throughout the study, participants and investigators who were involved in individual assessments were blinded to allocation. Two physiotherapists experienced in the GTT technique who were not involved in the evaluation of the individuals performed the GTT itself and the sham intervention (placebo) according to the randomization code.

### 2.6. Intervention

Intervention with GTT consisted of two sessions lasting between 40 and 45 min each. Sessions were performed at baseline (day 0) and after 45 days. We used this protocol of two 45 min sessions, as it is currently routinely performed in clinical practice and in previous research [53,54]. The treatments were performed only by two physiotherapists (CFTCS and MHT) with complete training in micro-physiotherapy.

During intervention with the patient lying on the stretcher, the experienced physiotherapist performed palpations of superficial and deep tissues, testing the physiological mobility of each tissue and the treatment sought to restore the mobility lost in cases of dysfunction [52]. Thus, the physical therapist used both hands to perform small, gentle approximation and separation movements in different regions of the participant’s body (Appendix A). Gentle touches were performed on the skin for the treatment of dysfunctional muscles, following a dermal map that contains the main embryological correspondences of the muscles in relation to their metamere of origin, Appendix A. In the first session, all mobility restrictions found in the epidermis were treated with a gentle touch and mobility in all parts of the body was restored until the physical therapist found no more restrictions. In the second session, new tissue restrictions were found in different parts of the body when compared to the regions found in the first session. Again, the treatment with gentle touch restored tissue mobility, being performed on all parts of the body until the physical therapist could no longer find restrictions [53,54]. The GTT was performed globally. The intervention protocol included basic treatment up to NP3 level (from the French, *Nouveau Perfectionnement*), as well as controls and corrections for advanced levels (P3, P4, P5 and P6, from the French, *Perfectionnement*) [52]. Due to the nature of the technique (manual therapy), it was not possible to blind the therapist, thus constituting a limitation of the study.

The placebo group was also subjected to two sessions where the patients underwent the same procedures, but with a simulated “sham GTT” approach—placebo—simulated for the same period of time, by the same physical therapists. During the sham treatment, gentle touches mimicking the technique were performed on random regions of the body, the dermal map of embryological correspondence of the muscles was not followed as performed in the intervention group [52].

### 2.7. Instruments and Assessments

#### 2.7.1. Primary Outcome Measures

##### VAS for Pain

The VAS for Pain was utilized to assess pain levels as the primary outcome. This tool is a widely used tool consisting of a horizontal line, 100mm in length, from 0 to 10. The value of 0 is equal to no pain while 10 is equal to the worst pain the patient has perceived. The rating is then measured from the left edge to the score marked by the patient (in millimeters). Participants were asked to answer the following question using the pain VAS: considering your pain, how intense was your worst pain during the last 24 h [58]? Participants’ pain levels were measured before interventions (Baseline) and on days 15, 45, 60, 90, 120 and 180 after treatments (Figure 1).

### 2.8. Secondary Outcome Measures

#### 2.8.1. Short-Form McGill Pain Questionnaire (MPQ-SF)

The MPQ-SF was used as a qualitative secondary outcome measure and is comprised of 15 descriptors divided into two subscales consisting of 11 sensory words and 4 affective words. These descriptors are subsequently rated on an intensity scale as 0 = none, 1= mild, 2 = moderate or 3 = severe. Three pain scores are derived from the sum of the intensity rank values of the words chosen for sensory, affective and total descriptors. The SF-MPQ also includes the Present Pain Intensity (PPI) index of the standard MPQ and a VAS. The present study used the validated Portuguese versions of the questionnaires [59]. Pain levels were assessed prior to any interventions (Baseline) and on days 15, 45, 60, 90, 120 and 180 after treatments (Figure 1).

#### 2.8.2. FIQ

FIQ: The Brazilian Portuguese version of the FM impact questionnaire was used to assess the impact of FM on health status [60]. The FIQ is comprised of assessments for 10 items: 1 assessment each for feel good, missed days of work, job difficulty, pain, fatigue, rested, stiffness, anxiety, depression and 10 subitems for physical function. The impact of fibromyalgia on participants was evaluated before interventions (Baseline) and on days 15, 45, 60, 90, 120 and 180 after treatments (Figure 1).

#### 2.8.3. SF-36 Quality of Life Questionnaire

The SF-36 is a multicultural scale consisting of 36 questions and categorized into eight domains: physical functioning (PF, 10 items), general health (GH, 5 items), physical role (i.e., role limitations due to the physical health problems, RP, 4 items), bodily pain (BP, 2 items), social functioning (SF, 2 items), vitality (VT, 4 items), emotional role (i.e., role limitations due to emotional problems, RE, 3 items), and mental health (MH, 5 items). For each domain, a score ranging from 0 to 100 was assessed with a higher score indicating better health. A Brazilian Portuguese version of the assessment was used [61]. The quality of life of participants was evaluated before interventions (Baseline) and on days 15, 45, 60, 90, 120 and 180 after treatments (Figure 1).

### 2.9. Blood Samples

Fasting samples were collected at baseline and 60 days after intervention (Figure 1) [62,63]. A 5 mL sample of peripheral blood was collected via venipuncture from the venous plexus in the antecubital fossa of the upper extremity in vacutainers (BD Vacutainer^®^ SST^®^ II) without additives and maintained for 20 min at room temperature. It was then centrifuged at 3000 rpm for 10 min at 4 °C, and immediately serum was collected, aliquoted and immediately stored at −80 °C until assay. Sampling was carried out between 8.00 and 9.00 a.m.

### 2.10. Serum Immune-Inflammatory Biomarkers Level

Sandwich Enzyme-Linked Immunosorbent Assay (ELISA) kits (DuoSet, R&D Systems, Minneapolis, MN, USA) were utilized to determine serum concentrations of BDNF, IL-8 and IL-10 in a volume of 100 µL, according to manufacturer’s instructions. The concentrations of interleukins and neurotrophic factor were estimated by interpolation of an 8-point standard curve with colorimetric measurements at 450 nm (corrected by subtracting readings at 540 nm) on a plate spectrophotometer (Perlong DNM-9602, Nanjing Perlove Medical Equipment Co., Nanjing, China). For the dosage of BDNF, the serum samples were diluted 1:3 and later multiplied by the dilution factor for the final result. All results were expressed in pg/mL [62,63].

### 2.11. Measurements of Neurotransmitters Levels

Catecholamine levels were determined by high-performance liquid chromatography (HPLC) and quantified by using electrochemical detection as previously described [64]. Serum samples were precipitated by the addition of one volume (1:1, *v*/*v*) of 0.5 M perchloric acid containing 0.02% sodium metabisulfite and centrifuged (16,000× *g* for 10 min at 4 °C). Monoamines and the metabolites present in the supernatants were assessed by HPLC (Alliance e2695, Waters, Milford, CT, USA), with electrochemical detection (Waters 2465, Waters, Milford, CT, USA) with 35 °C column temperature and the chromatographic run duration of 15 min. Twenty microliters of the supernatants were analyzed in a 150 × 2.0 mm, 4 µm, C18 column (Synergi Hydro, CA, USA) and an isocratic elution of 90 mM sodium phosphate, 50 mM citric acid, 2.3 mM sodium 1-heptane-sulfonate, 50 µM ethylenediaminetetraacetic acid, 10% acetonitrile, pH 3.0, with a flow of 0.20 mL/min. The results were expressed in η mol/L.

### 2.12. Measurements of BH4 Levels

BH4 levels were determined by HPLC and quantified using electrochemical detection as previously described with some modifications [38,65]. Urine samples were collected with a minimum of 2 h retention in a plastic cup. Urine samples were added to one volume (1:10, *v*/*v*) of 0.1 N HCl and immediately frozen and stored at −86 °C. Urine samples were thawed on ice and protected from light. Afterwards, samples were centrifuged (16,000× *g*; 10 min; 4 °C) and 20 μL of supernatant was transferred to an HPLC vial for analysis. The HPLC analysis of BH4 was carried out in an HPLC (Alliance e2695, Waters, Milford, CT, USA) by using a Waters Atlantis dC18 reverse phase column (4.6 × 250 mm; 5 μm particle), with a flow rate set at 0.7 mL/min and an isocratic elution of 6.5 mM NaH_2_PO_4_, 6 mM citric acid, 1 mM sodium octyl sulfate, 2.5 mM diethylenetriaminepentaacetic acid, 160 μM dithiothreitol and 8% acetonitrile, pH 3.0. The temperature of the column compartment was set at 35 °C. The identification and quantification of BH4 was performed by an electrochemical detector (module 2465, Waters, Milford, CT, USA) with a voltage of +450 mV. The results were expressed as µmol/mmol creatinine to urine. Creatinine levels were measured by using a standard commercial kit.

### 2.13. Statistical Analysis

The results were analyzed with the Graph Pad Prism program (version 6.0—La Jolla, CA, USA). Initially, the Shapiro–Wilk normality test was applied to evaluate the normality of the data with results expressed as means and standard deviations (SD). For the comparisons between pre and post values, repeated measures (RM) two-way ANOVA was utilized followed by the Bonferroni post hoc test. Data were converted to the area under the curve (AUC) using one-way ANOVA followed by Student Newman–Keuls test when appropriate [46]. When comparing fibromyalgia vs. healthy individual groups, the unpaired Student *t*-test was used for the parametric data or the Mann–Whitney test for the non-parametric data. A mixed ANCOVA model in which the independent variables were the time, the intervention (GTT and simulated intervention), interaction time vs. the intervention group and subject identification was used to analyze the score change on the VAS (0–10). The effect of interventions on the pain score was adjusted to depressive diagnosis, anxiety diagnosis, dopamine, serotonin, urinary tetrahydrobiopterin and the baseline serum BDNF. All analyses were corrected for multiple comparisons using the Bonferroni test. Standardized mean difference (SMD) ((post minus pre)/baseline standard deviation (SD)). Threshold values for Cohen’s effect size (ES) statistics were: small, 0.20; moderate, 0.50–0.60 and large, 0.80. All analyses were performed with two-tailed tests. A type I error of 5% was accepted. Statistical analyses were performed assuming intention-to-treat. The analyses were performed with the SPSS version 22.0 (SPSS, Chicago, IL, USA).

## 3. Results

Of the 136 female patients with a diagnosis of FM that were eligible for this study, 72 met the exclusion criteria resulting in 64 patients being selected for random allocation into one of the two groups: GTT (n = 32) or placebo (n = 32), as illustrated in Figure 2.

Baseline characteristics are presented in Table 1. The randomization produced balanced groups. The mean (SD) age of the participants was 53.7 (9.6) years for GTT and 53.2 (8.1) years for placebo. The body mass index (BMI) was 29.0 (5.1) kg/m² for the GTT and 28.9 (5.9) kg/m² for the placebo. The time of FM diagnosis was 7.3 (5.2) years for GTT and 6.9 (4.9) years for placebo. Differences between the groups with respect to pain, quality of life or serum biochemical analyses (Table 1) were not statistically significant.

### 3.1. Assessment of Pain: VAS and McGill-SF Questionnaire

VAS scores between the intervention and the placebo groups are not statistically significant as presented in Figure 3 and illustrated by the mean evaluations (Figure 3A). However, the intervention group presented a lower pain score (VAS) in the intragroup analysis in the 60-day period (*p* = 0.0456). The AUC of the VAS score of the intervetion group was lower (*p* = 0.0126) in comparison to the placebo group (Figure 3B).

Pain scores measured with the McGill-SF questionnaire did not reveal a statistical significance difference between the two groups in the sensory descriptor, affective descriptor, pain rating index (PRI) and present pain intensity (PPI) as illustrated by the mean evaluations throughout the study (Figure 4A,C,E,G). Interestingly, in the AUC analysis the intervention group had lower affective scores (*p* = 0.0416) when compared with the placebo group (Figure 4B).

### 3.2. Assessment of Quality of Life: FIQ and SF-36 Questionnaires

No statistically significant differences were found between groups in the FIQ and SF-36 Quality of Life Questionnaire. The results are illustrated in Appendix A, respectively.

### 3.3. Assessment of Inflammatory Mediators: Serum Neurotrophic Factor and Cytokines Levels

Figure 5 and Appendix A illustrate serum BDNF, IL-10 and IL-8 levels. Initially we observed that the FM group had higher levels of BDNF (*p* = 0.0425) when compared to the healthy individuals group (Appendix A). The FM group had lower levels of IL-10 (*p* = 0.0127) when compared to the healthy individuals group (Appendix A). No differences were found in serum IL-8 levels when compared to the healthy individuals group (Appendix A). We observed that the GTT group had lower levels of BDNF *(p* = 0.0214) when compared with the placebo group post intervention (Figure 5A). No differences were found in serum IL-10 (Figure 5B) and IL-8 levels (Figure 5C) when comparing GTT vs. placebo groups.

### 3.4. Assessment of Neurotransmitters or Neuromodulators: Serum Catecholamines, Monoamines and BH4 Levels

There was no statistically significant difference in dopamine and 5-HIAA levels in comparison between individuals with fibromyalgia and healthy individuals (Appendix A). The FM group had lower levels of serotonin (*p* = 0.0084) when compared to the healthy individuals group (Appendix A). No statistically significant differences were found in serum dopamine (Figure 6A), BH4 (Figure 6B), serotonin (Figure 6C) and 5-HIAA levels (Figure 6D) when comparing GTT vs. placebo groups.

### 3.5. GTT, Pain Relief and Neuroplasticity at Baseline in FM Syndrome: Exploratory Analysis

A mixed ANCOVA model in which the independent variables were the time, the intervention (GTT and placebo intervention), interaction time vs. the treatment group and subject identification was used to analyze the absolute variation on the score on the VAS at each time point of the follow-up. The cumulative mean in the VAS (0–10) across the time by paired comparison is presented in Figure 3.

An ANCOVA mixed model revealed a significant main influence of interventions on the VAS by the intervention influence (F = 14.39; *p* = 0.01). The analysis did not show a significant interaction between the group of interventions and time (F= 0.08; *p* = 0.93). The time influence was not observed (F = 0.58; *p* = 0.44). Higher baseline serum BDNF and 5-HIAA were related to a higher reduction in the pain score across time. In the same way, those with a history of anxiety disorder showed a higher reduction in pain scores across time. In contrast, higher serum dopamine at baseline showed a lower influence of intervention across time.

The change within the group was significant in both treatment groups (*p* < 0.001, for all comparisons). The mean (SD) on the VAS pretreatment vs. the cumulative pain scores according to cumulative means (T1 to T6) in the GTT was 7.74 (1.25) vs. 5.99 (2.55), respectively, a difference of 1.75 (2.19) (t = 7.95; *p* < 0.001). That is a treatment effect of a large effect size (Cohen’s f2 = 0.8). In contrast, in the placebo group it was 8.20 (7.20) vs. 7.20 (1.81), respectively, a difference of 1 (2.16) (t = 4.53; *p* < 0.001). That is a treatment effect of medium effect size (Cohen’s f2 = 0.49).

The mean (SD) of the VAS pretreatment vs. the cumulative pain scores according to cumulative means (T1 to T6) by marginal means in the GTT was 7.80 (1.30) vs. 5.99 (2.55), whereas in the placebo group it was 8.19 (1.30) vs. 7.20 (1.81). The effect based on the absolute mean change from baseline was −26.64% (30) in GTT and −10.04% (27) in the placebo treatment, with an absolute mean difference of −16.60%. It was an effect of moderate effect size (Cohen’s f2 = 0.58), as illustrated in Table 2.

## 4. Discussion

To the best of our knowledge, this is the first randomized trial that compared two sessions of GTT 45 days apart with placebo intervention in women with FM. Our results suggest a reduction in pain but did not produce changes in the quality of life of these individuals. Furthermore, the lowest pain score observed in the GTT group is related to the neuroplasticity state at baseline. The most relevant additional findings of the present work are that (i) individuals with FM who had higher serum levels of BDNF showed a higher reduction in pain scores at the follow-up despite the intervention. However, the GTT intervention reduced the serum level of BDNF, in contrast this reduction was not observed in the placebo group; (ii) lower serum levels of IL-10 and serotonin in individuals with FM when compared to healthy individuals; (iii) there were no significant differences in the serum levels of IL-8, 5-HIAA, dopamine or in the urinary levels of BH4 when comparing individuals with FM with healthy individuals; (iv) there were no significant differences in serum levels of IL-8, IL-10, serotonin, 5-HIAA, dopamine or in urinary BH4 levels between the GTT and placebo groups; (v) higher baseline serum BDNF and 5-HIAA or those with a history of anxiety disorder showed a higher reduction in pain scores across time. In contrast, higher serum dopamine at baseline was associated with a lower effect of intervention across time.

In fact, evidence-based guidelines emphasize the need for multimodal treatments in FM cases [66]. Although the physical intervention of musculoskeletal disorders is undergoing a transformation, we have to recognize that hands-on techniques are a central element of physical therapy identity [67] and are appreciated and expected by patients in the context of a therapeutic ritual historically based on the use of touch [68]. Even though we are mature enough to accept a transformation of our heritage as a sign of professional development, we have now recognized that pain modulation caused by hands-on techniques plays a role in many patients beyond basic biomechanics, which represents only one of the possible explanations of this neurobiological effect [69,70]. Additionally, theoretical models have been described presenting neurophysiological (e.g., pain neural networks modulation) and context-related (e.g., placebo) components that currently better explain the effects of touch (mechanical stimulation) on pain modulation [71,72].

The results showed that the intervention group had lower pain scores compared with the placebo group when evaluated by the VAS for pain. Its impact on pain measures agrees with a set of previous studies of gentle touch therapies. It has been demonstrated that gentle human touch techniques alleviate pain and improve physiological parameters in preterm infants during heel lancing [73]. Furthermore, Weze et al. (2003) carried out an uncontrolled study utilizing gentle touch in individuals with cancer at the Center for Complementary Care in Eskdale, Cumbria. They observed a significant improvement in psychological and physical functioning, with positive effects on quality of life. The most improvements have been seen in ratings for stress and relaxation, depression/anxiety and severe pain/discomfort, which are also evaluated by the VAS [74].

Specifically with micro-physiotherapy, which was the gentle touch technique employed in this study, Baconnier et al. (2016) carried out a double-blind randomized controlled clinical trial to assess the effectiveness of micro-physiotherapy in post-traumatic neck pain [53]. They observed that only the group treated with micro-physiotherapy had pain reduction and increased range of neck flexion and extension. In the control group, significant improvement was not observed. In addition, studies have been performed with micro-physiotherapy in the treatment of visceral disorders. Grosjean et al. (2017) analyzed the effects of micro-physiotherapy on the severity of symptoms in patients with irritable bowel syndrome (IBS) [54]. They recruited 61 patients that were randomly distributed into two groups, a control and a micro-physiotherapy group. The control group received a false procedure. The severity of symptoms of patients with IBS was assessed by a gastroenterologist. The results showed that patients who received the first session of micro-physiotherapy had a 74% improvement in IBS symptoms, while patients in the control group showed only a 38% improvement in IBS symptoms, thus having a statistical difference in improvement between the groups. It was observed that after the second intervention session, the effects of both groups were maintained. Thus, the authors concluded and suggested that micro-physiotherapy significantly improves the symptoms of IBS and that its use in primary health care should be further explored.

Additionally, it has been observed that the intervention group had a lower affective pain score compared with the placebo group when assessed by the MPQ-SF for pain.

Considering the present data, we may speculate that the analgesic action of GTT is probably linked to an activation of C-tactile fibers caused by the gentle touch that makes up this technique. Enhanced activity of C-tactile afferents (CT afferents) may induce a ‘limbic touch’ response, resulting in emotional and hormonal reactions [75,76,77]. The feeling of calm and well-being produced for therapies that use gentle touch may be associated with the effect of the gentle stimulation of manual touch mediated by these neuronal fibers [78,79]. This mechanism of action makes them ideal to treat pathologies that present alterations in the stress axis, which is the case of FM [7,78,79]. However, further research is needed to support this hypothesis.

Concerning the mechanism through which GTT exerts its analgesic action, our findings highlight that the lowest pain score observed in the GTT group compared to the placebo group is correlated with the neuroplasticity state, according to serum BDNF [80]. This conclusion derives from the fact that the GTT intervention resulted in significant serum BDNF level reduction in individuals with FM relative to placebo, followed by a decrease in pain scores. The direction of modulation depends on the state of the neural networks involved in pain processing [81] and that baseline neuroplasticity may be a determinant of interindividual variability of the lowest pain score observed in the GTT group compared to the placebo group. It has been suggested that biomarkers related to neuronal plasticity such as BDNF are increased in FM and other central sensitivity syndromes [82]. Baseline BDNF levels in FM patients approximate levels seen in individuals in a study of patients with central sensitivity syndrome characterized by persistent somatic or visceral nociception [82], comorbidities which are present in FM [7]. Despite elevated levels of circulating BDNF in patients with FM compared to matched controls [83], some studies have found no significant differences [84]. Nevertheless, our results replicated those in a similar study in which BDNF levels decreased after attachment-based compassion therapy, with significant pre-post intervention decreases in BDNF accompanied by benefits in the general health status [85].

Regarding the lowest pain score observed in the GTT group compared to the placebo group with respect to the BDNF levels, we also believe that the lowest in serum BDNF levels is related to the role of this mediator in pain processing at both peripheral and perhaps spinal levels [86] and mainly from the action of C-tactile fibers in reducing pain (this processing) [87,88] and that due to the body regions and gentle nature of the touches performed during the micro-physiotherapy session but not in the placebo group, C-tactile fibers are effectively stimulated. The BDNF is expressed in the adult by TrkA-positive primary sensory neurons [89], most of which are nociceptive C fibers. It is anterogradely transported to the central terminals of these afferents in the superficial laminae of the spinal dorsal horn [90,91,92]. BDNF in the dorsal horn could have three roles. It might contribute to the transfer of information related to the intensity and duration of noxious stimuli from primary afferent to second-order neurons, i.e., determine basal pain sensitivity. Alternatively, it may be involved in the stimulus-induced activity-dependent plasticity of somatosensory pathways, central sensitization [93]. Finally, BDNF may have a role in generating inflammatory pain hypersensitivity. In humans, slow and gentle touch is supposed to ease pain [94]. Recently, a modulation of laser-evoked pain by CT-targeted touch on the contralateral extremity has been reported. The effect of CT-targeted touch (attenuating/increasing pain) was dependent on the attachment styles (attachment anxiety/attachment avoidance) of the participants [95]. Based on the theoretical framework presented, we support the idea that the probable activation of CT fibers by micro-physiotherapy, but not in the placebo group, can reduce pain transmission and thus the BDNF level is an important mediator involved in this process.

A chronic pro-inflammatory state in FM has been attributed to a variety of physical and/or psychological stressors with subsequent direct impacts on central pain processing [96]. It is theorized that alterations in these pain pathways may involve low-grade chronic basal neuroinflammation processes, with stress peptides triggering the release of neurosensitizing mediators [97]. Elevated levels of pro-inflammatory cytokines (e.g., IL-6, IL-8, and tumor necrosis factor (TNF)) and suppression of anti-inflammatory cytokines (e.g., IL-10) have been highlighted and observed as a proposed of inflammatory mechanism in the development of FM [98,99].

In the present study, we found that IL-10 levels were decreased in patients with FM, which corroborates a previous study [99], although elevated levels have also been noted in individuals with FM [28]. Reduction in nociceptor sensitization has been demonstrated by IL-10’s anti-inflammatory properties [100], which may explain the pain experienced by patients. However, no difference was delineated in levels of IL-10 between intervention groups.

Another cytokine that has been widely studied in FM is IL-8 [5,28,101] and its contribution to pain modulation [102]. Microglial cells and astrocytes synthesize IL-8 and have a pronounced role in the mediation of intercellular communication between glia and neurons through rapid alteration of the neuronal excitability [102]. Previous studies, including a systematic review and a meta-analysis, demonstrated increased levels of IL-8 in individuals with FM compared with non-affected individuals. In the present study, we found serum levels of IL-8 (5.8 pg/mL) similar to that found by other authors (6.0 pg/mL) [32]. However, there was no difference between the groups of FM and healthy individuals, corroborating the study by Kutlu et al. (2019) [103]. Serum IL-8 concentrations show great variability and researchers have associated this difference with the inclusion criteria and the method of analysis, as reviewed by Uçeyler et al. (2011) [35]. Since no statistical difference was found between healthy and FM individuals in the present study, there was no statistical difference between the groups of individuals with FM and those treated with GTT or placebo. This finding might be justified by the fact that the gentle touch used in the micro-physiotherapy technique stimulates the body to reestablish homeostasis [75,76,78,79]. Thus, it might be hypothesized that as the serum concentrations of IL-8 are similar between healthy and sick individuals, all individuals in the research were already in homeostasis in relation to the serum concentration of IL-8 and that this inflammatory mediator probably was not the main substance mediating FM symptoms in these patients [5,28,75,76,77,78,79].

Another relevant factor that might contribute to the development of FM is the altered levels of neurotransmitters [6,104]. These neurochemical imbalances in patients with FM have been associated with the dysregulation of pain perception characterized by allodynia and hyperalgesia [105,106]. However, previous research has shown conflicting results as to serum/plasma levels of catecholamines (dopamine, norepinephrine, and epinephrine) and indoleamines in patients with FM. Levels of dopamine have been reported to be increased, decreased and unchanged in subjects with FM compared to healthy controls [14,106]. Similarly, some authors have reported reduced levels of 5-hydroxytryptamine (5-HT) in FM compared with the control group [107], while others did not find differences between patients and healthy subjects [108].

In the present study we evaluated the levels of a catecholamine (dopamine) and an indolamine (serotonin) and their main metabolite (5-HIAA). Dopamine and 5-HIAA levels were unaltered in FM subjects compared to healthy subjects. We also found no significant difference in dopamine, serotonin and 5-HIAA levels between FM subjects undergoing GTT or placebo. However, we found lower serum serotonin levels in subjects with FM compared to healthy controls. Thus, the results of the present study corroborate the literature by observing lower serum serotonin levels in subjects with FM [99]. In the pathophysiology of FM, variation in 5-HT levels results in several aspects, including imbalance of the hypothalamic–pituitary axis [109], reduction in descending pain pathways related to pain inhibition, autonomic dysfunction, sleep disorders, anxiety and Raynaud’s phenomenon among others [7]. Furthermore, the positive effect of available pharmacological agents that increase serotonin levels in FM supports the importance of this indolamine for the pathogenesis of FM [110].

Finally, our results showed that GTT intervention did not alter the quality of life of women with FM. However, these findings were not surprising, as pharmacological agents widely used in FM management, such as the serotonin and noradrenaline reuptake inhibitors (SNRIs) duloxetine and milnacipran, did not provide any clinically relevant benefit over the placebo in improving health-related quality of life, reducing fatigue or reducing sleep problems [110].

Our findings should be interpreted with regard to some limitations of the present study. The intervention with GTT was carried out with only two sessions, as is routinely done in clinical offices. Perhaps more sessions could yield more benefits, such as improved quality of life. Furthermore, the lack of other Th-1 cytokines (including interferon-γ and TNF), Th-2 cytokines (including eotaxin and IL-6) and T regulatory cytokines (including transforming growth factor-β1) [28,111] does not allow us to describe in detail the status of the inflammatory response. This would allow us to compute more comprehensive inflammatory versus regulatory immunity regulation [28,111]. In addition, salivary cortisol levels could have been measured to analyze the influence of GTT more directly on the stress axis. Finally, the level of leisure activity of the individuals in the study was not evaluated and may have influenced the results of the dosages of catecholamines and indoleamines.

## 5. Conclusions

In conclusion, a lower pain score was observed in the GTT group compared to the placebo group without altering the quality of life in women with FM. Additionally, they suggest that serum BDNF at baseline predicted the impact of the intervention on pain measures across the treatment, under the conditions described in this study. In this sense, the present study encourages the use of these GTT techniques as an integrative and complementary treatment of FM.

## Figures and Tables

**Figure 1 jcm-11-04898-f001:**
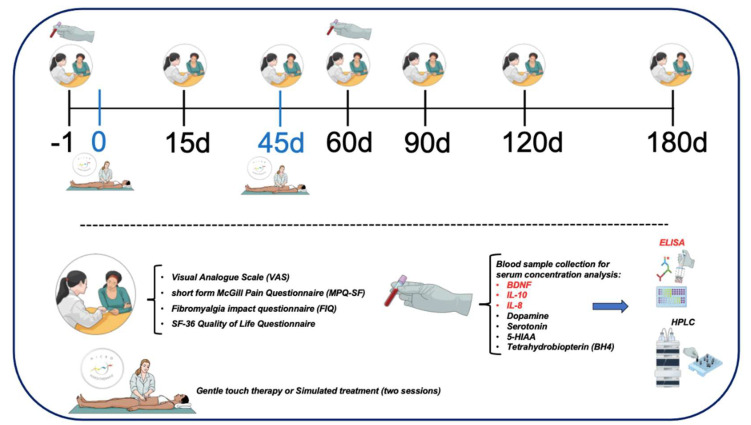
Timeline of intervention and analyses. ELISA: Enzyme-Linked Immunosorbent Assay; HPLC: high performance liquid chromatography; d: day.

**Figure 2 jcm-11-04898-f002:**
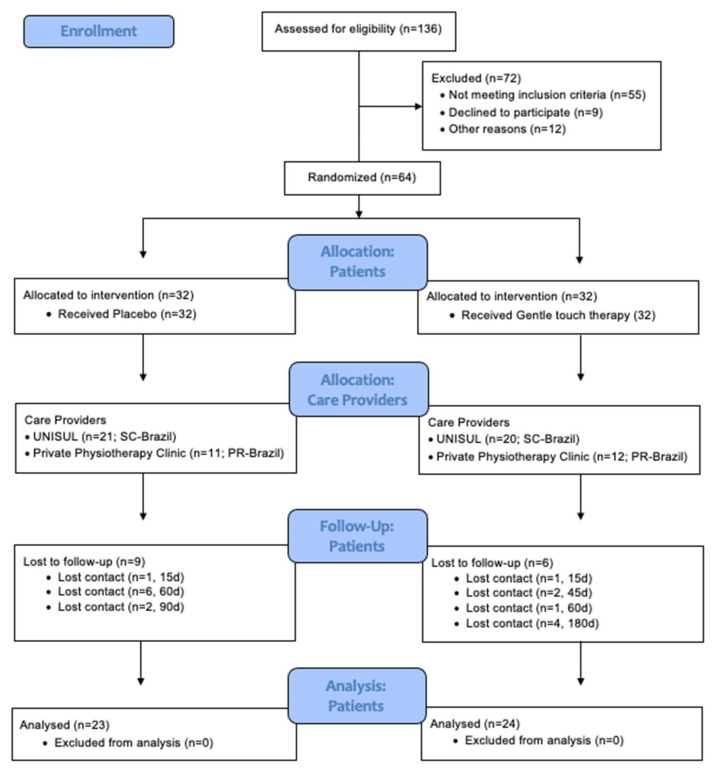
Flow diagram with timeline of the study according to the Consolidated Standards of Reporting Trials (CONSORT) of Statement for Randomized Trials of Nonpharmacologic Treatments. UNISUL: University of Southern Santa Catarina; PR: Paraná.

**Figure 3 jcm-11-04898-f003:**
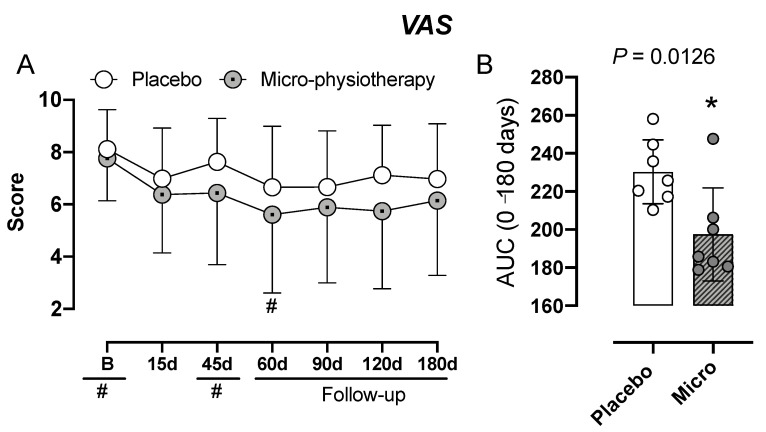
Pain assessed through the visual analogue scale (VAS). Panel (**A**) shows the evaluations at different times. Repeated-measures two-way analysis of variance followed by Bonferroni post hoc test. In panel (**B**) the result is expressed by the area under the curve (AUC). Unpaired Student *t*-test. # *p* < 0.05 when compared to GTT group in Baseline (**B**). * *p* = 0.0126 when compared to placebo group. GTT: gentle touch therapy group; d: day.

**Figure 4 jcm-11-04898-f004:**
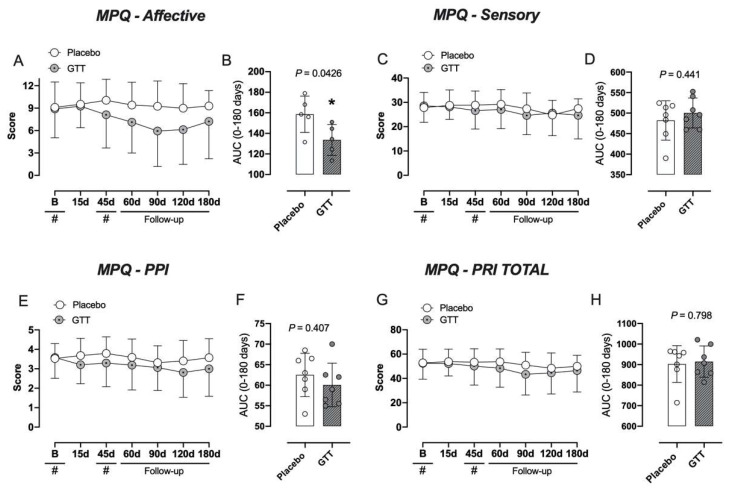
Pain assessed through the McGill Pain Questionnaire (MPQ). Panels (**A**,**C**,**E**,**G**) show the evaluations at different times. Repeated-measures two-way analysis of variance followed by Bonferroni post hoc test. In panels (**B**,**D**,**F**,**H**) the result is expressed by the area under the curve (AUC). Unpaired Student *t*-test was used for the parametric data or the Mann–Whitney test for the nonparametric data. # *p* < 0.05, * *p* < 0.05 when compared to placebo group. PRI: Pain Rating index; PPI: present pain intensity; GTT: gentle touch therapy group; d: day.

**Figure 5 jcm-11-04898-f005:**
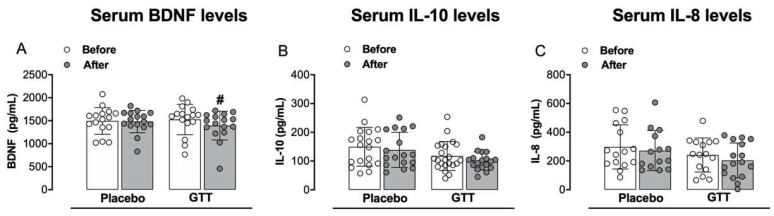
Assessment of serum levels of brain-derived neurotrophic factor (BDNF, panel (**A**)), interleukin-10 (panel (**B**)) and interleukin-8 (panel (**C**)). Repeated-measures two-way analysis of variance followed by Bonferroni post hoc test. # *p* < 0.05 when compared to placebo group. Before: baseline; After: after 60 days.

**Figure 6 jcm-11-04898-f006:**
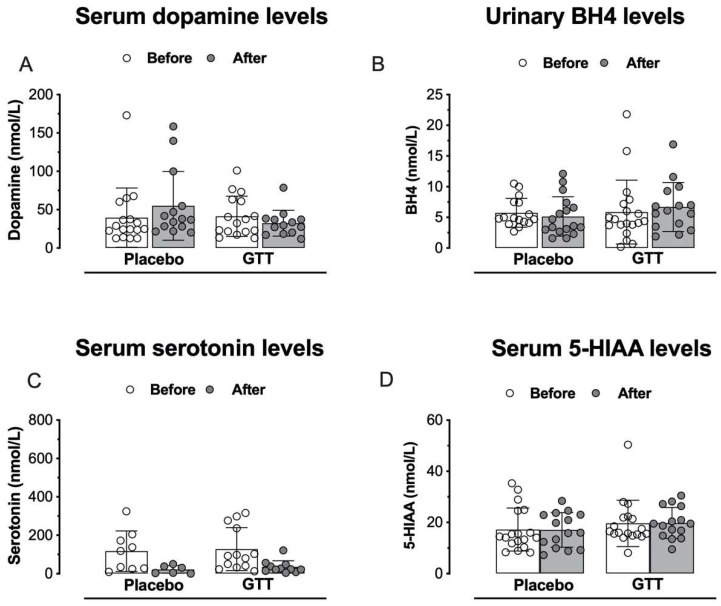
Assessment of serum levels of dopamine (panel (**A**)), serotonin (panel (**C**)), 5-hydroxyindolacetic acid (5-HIAA, panel (**D**)) and urinary level of tetrahydrobiopterin (BH4, panel (**B**)). Repeated-measures two-way analysis of variance followed by Bonferroni post hoc test. Before: baseline; After: after 60 days.

**Table 1 jcm-11-04898-t001:** Epidemiological and clinical characteristics at baseline, according to the intervention group, values are given as the mean (SD) or frequency (n = 64).

Characteristics	Gentle Touch Therapy (n = 32)	Placebo (n = 32)	*p*-Value
Age (years)	53.7 (9.6)	53.2 (8.16	0.837
Body mass index (Kg/m^2^)	29.0 (5.1)	28.9 (5.9)	0.984
Time of fibromyalgia diagnosis (years)	7.3 (5.2)	6.9 (4.9)	0.638
Physical activity (yes/no)	15/17	16/19	-
Smoking (yes/no)	4/28	6/26	-
Analgesic medication in use (yes/no)	15/16	15/17	-
History of anxiety (yes/no)	4/28	6/26	-
History of depression disorders (yes/no)	10/22	14/18	-
Drug active on the nervous system in use (yes/no) **	14/18	13/19	-
Selective Serotonin Reuptake Inhibitor in use (yes/no)	12/20	12/20	-
History of chronic disease (yes/no)	14/18	22/10	-
Hypertension (yes/no)	10/21	13/19	-
Type 2 Diabetes Mellitus (yes/no)	3/28	4/28	-
Pain on the VAS (range 0–10)	7.7 (1.6)	8.1 (1.5)	0.368
Short-form McGill Pain Questionnaire	52.4 (13.0)	52.7 (11.2)	0.926
Fibromyalgia Impact Questionnaire (FIQ)	61.6 (10.9)	65.3 (7.4)	0.995
SF-36 Quality of Life Questionnaire			
Physical functioning	30.1 (16.5)	35.7 (20.7)	0.237
General health	44.5 (17.3)	45.2 (22.5)	0.901
Physical Role	11.9 (22.8)	13.2 (25.3)	0.820
Bodily pain	25.4 (16.4)	24.3 (13.8)	0.767
Social functioning	42.1 (22.5)	44.7 (25.1)	0.738
Vitality	29.0 (16.1)	28.1 (17.1)	0.875
Emotional Role	19.8 (30.2)	28.1 (38.8)	0.343
Mental health	41.0 (20.4)	38.75 (22.4)	0.668
Serum BDNF (pg/mL)	1526 (329.8)	1496 (290.2)	0.788
Serum IL-8 (pg/mL)	4.8 (2.3)	5.9 (3.0)	0.270
Serum IL-10 (pg/mL)	2.3 (1.0)	2.9 (1.3)	0.090
Serum Serotonin (nmol/L)	127.6 (112.6)	117.0 (105.6)	0.825
Serum 5-HIAA (nmol/L)	19.6 (9.0)	17.2 (8.3)	0.409
Serum Dopamine (nmol/L)	41.2 (26.3)	39.6 (38.6)	0.887
Serum BH4 (nmol/L)	5.8 (5.1)	5.7 (2.3)	0.932

** Some patients were using more than one type of drug.

**Table 2 jcm-11-04898-t002:** Absolute mean changes in the visual analogue scale between intervention groups: mean difference with the confidence interval (95% CI) (n = 37).

Dependent Variable: Percentage of Change on Pain Scores by Visual Analogue Scale from Baseline						
	Beta	Std. Error	df	t	Sig.	CI 95%
Intercept	−58.4104	18.096799	147.489	−3.228	0.002	(−94.17 to −22.64)
Gentle touch therapy	−16.5963	4.374591	156.229	−3.794	0.000	(−25.23 to −7.95)
Placebo intervention	Reference 0					
Brain-derived neurotrophic factor (BDNF)	0.020730	0.009095	154.889	2.279	0.024	(0.004 to 0.04)
Dopamine	−0.152667	0.065176	155.827	−2.342	0.020	(−0.28 to −0.03)
Serotonin	−0.024950	0.024901	154.004	−1.002	0.318	(−0.07 to 0.02)
5-hydroxyindolacetic acid (5-HIAA)	0.880112	0.273803	156.154	3.214	0.002	(0.34 to 1.42)
Urinary tetrahydrobiopterin (BH4)	−0.272308	0.302144	126.154	−0.901	0.369	(−0.87 to 0.33)
History of depression diagnosis	0.176808	5.146371	156.334	0.034	0.973	(−9.98 to 10.34)
History of anxiety diagnosis	18.5303	7.145711	156.215	2.593	0.010	(4.42 to 32.64)

## Data Availability

The data that support the findings of this study are available from the corresponding author upon reasonable request.

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
