# Peer review of "Gentle Touch Therapy, Pain Relief and Neuroplasticity at Baseline in Fibromyalgia Syndrome: A Randomized, Multicenter Trial with Six-Month Follow-Up"

_jcm, 2022, doi:10.3390/jcm11164898_

Round 1
Reviewer 1 Report
“The Long-Term Effect of Micro-Physiotherapy in the Relief Pain on Fibromyalgia is Related to Neuroplasticity State at Baseline: A Randomized, Multicenter, Double-Blind, Placebo-Controlled Trial with Six-Month Follow-Up” is a very interesting study evaluating the efficacy of Micro-physiotherapy as an alternative therapy for relief of fibromyalgia symptoms, however, I consider that some details could improve the quality of the work before being published.
First, please clarify in the whole document the most significant results since in some sections an improved patient’s quality of life is mentioned besides a pain improvement, but in the majority of the results, it is indicated that the quality of life was not improved.
Introduction
Besides the information regarding Manual therapy (MT) included in this section, please add a little more detailed explanation in order to clearly point out what Manual therapy is. This would help to better understand the importance of this therapy and maybe promote it among patients.
The above also applies to Micro-physiotherapy. Please add information about the Micro-physiotherapy advantages over other types of Manual therapy.
Please explain why only two 45 minutes-sessions were selected as sufficient to effectively alleviate pain and improve the quality of life of FM patients. Include bibliographic references for this explanation.
Materials and Methods
Please explain the real difference between the Micro-physiotherapy group and the placebo group, i.e., what did the placebo group receive? Knowing these characteristics provides a broader view for the interpretation of the results.
Please justify why it was determined as necessary a 3 months-follow up for FM patients.
Taking into account the information included in the Introduction, please explain why different innocuous and not innocuous stimuli of different nature were not considered to be evaluated in order to assess allodynia and hyperalgesia in FM patients. Perhaps, Micro-physiotherapy could improve the response to certain stimuli to which patients are exposed in their environment.
In section 2.11, please indicate the temperature of the column and the chromatographic run duration.
Please justify why a healthy patient’s control group was not included.
Please emphasize that Cohen’s D test was used to know the effect size of treatments.
Results
Table 1: why the total of surveyed people does not sum up the 32 participants of each group? (Placebo; Smoking (yes/no)= 5/22), or (Placebo; Physical activity (yes/no) = 12/16)).
Figure 5: why a comparison between the ‘before group’ and the ‘after group’ of the Micro-physiotherapy effect was not performed?
Regarding lines 433 “Higher baseline serum BDNF and 5-HIAA were related to a higher reduction in the pain score across time” and 436 “Higher serum dopamine at baseline showed a lower effect of intervention across the time”, how did the authors know that baseline values of BNDF, dopamine, and 5-HIAA were high? Were they compared? Against what? Inside the group, some patients had higher levels than their fellows?
Figure 6: C and D panels letters are missing. Besides, is there any significant difference in panel D to be marked?
Discussion
All my comments could be supplemented with preclinical articles if little or no information on patients with chronic pain is not available.
What occurs in the Micro-physiotherapy group that does not occur in the placebo group to decrease BDNF levels in FM patients?
It is advisable to explain how the C fibers activation turns on the neuroimmune axis to regulate BDNF levels.
Please discuss if BDNF levels modify 5-HIAA levels. According to your results, it seems that actually they do, however, this is not discussed. This idea could be reinforced with the beneficial effects found in the MPQ-Affective.
Regarding line 540: “In the present study we found that IL-10 levels were decreased in patients with FM, which corroborates a previous study”, this is not shown in the Results section, nor a mention in the text nor plots show these significant differences. To demonstrate this, a healthy control group should be included.
Please comment on what you expect with more Micro-physiotherapy sessions and what would happen if such sessions are performed with less than 45 days between them.
Reviewer 2 Report
The manuscript describes the long-term effect of micro-physiotherapy in the relief pain on fibromyalgia. The major finding is a reduction in pain with no effect on the quality of life, and a reduction in BDNF serum level after treatment. It is a randomized, double-blinded, multicentred, parallel-group trial compared 1 intervention.
Micro-physiotherapy is a manual physiotherapy technique, where a light manual stimulation acts to stimulate self-healing. It is an alternative medicine technique proposed in a holistic approach for chronic pain diseases such as IBS.
The major issue is the technique and the lack of a real the sham control, and this problem is strictly correlated to the methodology itself. If the therapist “stimulates” the self-healing, he/she can non give a sham control.
The technique and the manual procedures performed on the patients are for their nature holistic and therefore not methodologically controllable.
In addition, to my knowledge it is not scientifically based that the etiology of pain in FM could be identified by micropalpation. The bases of the proposed treatment lay on a non-conventional approach and the described technique cannot be reproduced.
In particular:
- -The duration of the intervention may span from 30 to 45 min (± 50% !)
- - Line 230 “the experienced physiotherapist performed micropalpations, identifying the etiology of the patients' symptoms and encouraging self-healing”
Reviewer 3 Report
Dear authors,
Congratulations on the article, the time spent and the research carried out.
My comments are very positive. You have a good conceptualisation of the data, there is an exhaustive search carried out and also, the method and procedure are very rigorous.
My suggestions for improvement are:
-At the end of the conclusion, indicate the possible theoretical implications for the scientific community and what this means for the subject and researchers researching fibromyalgia.
-At the end of the conclusion, indicate possible practical implications for society.
-Indicate the strength and limitations of the study, as well as any future ideas or possible lines of research.
Best regards,
Round 2
Reviewer 1 Report
This is the second revision of the work: “The long-term effect of gentle touch therapy in the relief pain on fibromyalgia is related to neuroplasticity state at baseline: A randomized, multicenter, double-blind, placebo-controlled trial with six-month follow-up”.
Congratulations to the authors. This version is of better quality than the previous one. As a reviewer, I request the following changes:
1. Answer: In this research, we used this protocol of only two 45 minutes-sessions, as it is currently routinely performed in clinical practice and in previous research (Grosjean et al., 2017; Baconnier et al., 2016).
Please, add the above comment and the references to the manuscript in section 2.6
2. Answer: We thank the referee for this comment and suggestion. We believe that due to the body regions and gentle nature of the touches performed during the microphysiotherapy session but not in the placebo group, C-tactile fibers are effectively stimulated. We also believe that the reduction in serum BDNF levels is related to the role of this mediator in pain processing at both peripheral and perhaps spinal levels (Mannion et al., 1999) and mainly from the action of C- tactile fibers in reducing pain (this processing) (Habig et al., 2017; Case et al., 2016). As supported by the theoretical framework below: The neurotrophin brain-derived neurotrophic factor (BDNF) is expressed in the adult by TrkA-positive primary sensory neurons (Apfel et al., 1996), most of which are nociceptive C fibers (No C-tactile fibers). It is anterogradely transported to the central terminals of these afferents in the superficial laminae of the spinal dorsal horn (Yan et al. 1997; Zhou et al., 1996; Michael et al. 1997), where it is localized in dense core synaptic vesicles (Michael et al. 1997). BDNF in the brain regulates synaptic efficacy (Korte et al., 1995; Kang et al., 1996) and may, if released from afferent terminals in the spinal cord, act as synaptic modulator either presynaptically on primary afferent TrkB-positive terminals or postsynaptically on TrkB- expressing neurons in the spinal cord. BDNF in the dorsal horn could have three roles. It might contribute to the transfer of information related to the intensity and duration of noxious stimuli from primary afferent to second-order neurons, i.e., determine basal pain sensitivity. Alternatively, it may be involved in the stimulus-induced activitydependent plasticity of somatosensory pathways, central sensitization (Woolf, 1983). Central sensitization is a C fiber-mediated increase in excitability of dorsal horn neurons that generates hypersensitivity by recruiting previously subthreshold mechanoreceptors inputs to pain transmission pathways (Torebjork et al., 1992). Finally, BDNF may have a role in generating inflammatory pain hypersensitivity. Human, skin contains a subgroup of C-fibers, the C-low threshold mechanoreceptive afferents ((C-LTMR) C-tactile or C-touch (CT) fibers) that are linked with the signaling of affective aspects of human touch. In mice, the cell bodies of the C-LTMRs are randomly distributed in the dorsal root ganglia, CLTMRs terminate in Lamina II of the dorsal horn accompanied by Aδ and Aβ afferents [Li et al., 2011]. They enter the lamina I/II spinothalamic pathway up to the ventromedial posterior thalamic nucleus [Craig, 2022; Andrew, 2010). In humans, the main brain areas receiving C-LTMR information belong to the somatosensory system and affect processing brain networks like the contralateral posterior insular cortex [Olausson et al., 2002] or the medial prefrontal cortex [Liljencrantz et al., 2013). The intensity of CT targeted touch is encoded in the primary and secondary somatosensory cortex (S1 contralateral, S2 bilateral), whereas the pleasantness is encoded in the pregenual anterior cingulate cortex [Case et al., 2016]. C-LTMRs also activate regions involved in reward processing (putamen and orbitofrontal cortex [Sailer et al., 2016;Olausson et al., 2002]) and in processing of social stimuli (posterior superior temporal sulcus [Voos et al., 2013; Bennett et al., 2013; Gordon et al., 2013). Animal experiments indicate that CT activation reduces pain, but the precise anatomical localization of this phenomenon is still unknown [Le Bars, 2002; Lu and Perl, 2003]. In humans, slow and gentle touch is supposed to ease pain [Mancini et al., 2014]. Recently a modulation of laser evoked pain by CT targeted touch on the contralateral extremity has been reported. The effect of CT targeted touch (attenuating/ increasing pain) was dependent on the attachment styles (attachment anxiety/ attachment avoidance) of the participants [Krahee et al., 2016]. Based on the theoretical framework presented, we support the idea that the probable activation of CT fibers by microphysiotherapy, but not in the placebo group, can reduce pain transmission and thus BDNF, which is an important neurotransmitter involved in this process.
Please, include a summary of the above explanation and the references in the discussion section as it explains the beneficial effects of the gentle touch therapy with respect to the BDNF levels.
3. Answer: We thank the referee for this comment and suggestion. Yes, some individuals had higher levels than others, however, we do not have a reference value to define what is high. For this statistical analysis we used the mixed ANCOVA model which the independent variable was the time, the intervention (micro-physiotherapy and simulated intervention), interaction time vs. the intervention group, and subject identification were used to analyze the score change on the VAS (0-10). The effect of interventions on the pain score was adjusted for depressive diagnosis, anxiety diagnosis, dopamine, serotonin and the baseline serum BDNF. All analyzes were corrected for multiple comparisons using the Bonferroni test. Standardized mean difference (SMD) [(post minus pre)/baseline standard deviation (SD)].
In further investigations, it would be interesting to analyze subgroups of patients with FM according to their BDNF, 5-HIAA, and dopamine levels since it looks like the gentle touch therapy has a greater beneficial effect in patients presenting these characteristics, and that could be such a biomarker for the type of patient in which a greater beneficial effect of the gentle touch therapy would be expected.
Reviewer 2 Report
The reviewer appreciated the author’s efforts to explain the technique.
GTT is described as treatment which relays on the therapist’s identification and treatment of altered mobility of soft tissue. The lack of a real sham control and therefore blind treatment, due to the nature of the technique, result the major obstacle to publication.
